# Metallization by Sputtering to Improve the Bond Strength between Zirconia Ceramics and Resin Cements

**DOI:** 10.3390/jfb12040062

**Published:** 2021-11-17

**Authors:** Tatsuya Kimura, Yujin Aoyagi, Norimasa Taka, Mitsugu Kanatani, Katsumi Uoshima

**Affiliations:** 1Division of Bio-Prosthodontics, Faculty of Dentistry, Graduate School of Medical and Dental Sciences, Niigata University, 2-5274 Gakkocho-dori, Chuo-ku, Niigata 951-8514, Japan; t.kimura@dent.niigata-u.ac.jp (T.K.); fish@dent.niigata-u.ac.jp (K.U.); 2Division of Preventive Dentistry, Faculty of Dentistry, Graduate School of Medical and Dental Sciences, Niigata University, 2-5274 Gakkocho-dori, Chuo-ku, Niigata 951-8514, Japan; taka@dent.niigata-u.ac.jp; 3Division of Biomimetics, Faculty of Dentistry, Graduate School of Medical and Dental Sciences, Niigata University, 2-5274 Gakkocho-dori, Chuo-ku, Niigata 951-8514, Japan; kanatani@dent.niigata-u.ac.jp

**Keywords:** zirconia, surface treatment, metallization, titanium sputtering, bonding strength

## Abstract

Zirconia has been used as a prosthesis material for over a decade because of its excellent mechanical properties and esthetics. The surface treatment for zirconia generally involves sandblasting and the application of primers for favorable bond strength between the surface and resin. However, sandblasting causes the microcracking and chipping of the zirconia surface. To overcome these challenges, the metallization of the zirconia surface was performed. Ti and Au were sputtered on yttria stabilized zirconia (YSZ) disks and heated to 800 °C for 15 min in air. These disks were bonded to stainless-steel rods using resin cement. Then, shear bond strength tests were performed using an Instron-type testing machine. The shear bond strength of the Ti sputtering group was significantly higher than that of the other groups. According to the results of X-ray photoelectron spectroscopy and electron probe microanalysis, the Ti-sputtered YSZ surface contained both sub-titanium oxide and titanium oxide before heating. Sub-titanium oxide was converted to titanium oxide by heating. These results suggest that metallization using Ti is effective for zirconia surface treatment to improve the shear bond strength between YSZ and resin cement. This metallization technique for YSZ has potential in clinical applications.

## 1. Introduction

Owing to the increasing esthetic requirements of dental patients, dental treatments using ceramic crowns have been gaining popularity [1,2,3,4,5,6]. Zirconia has been used as a material for dental prostheses for over a decade because of its excellent mechanical properties and esthetics [2,3,4,5,6,7,8,9]. The most commonly used zirconia ceramic is a tetragonal zirconia polycrystalline stabilized with 3 to 8 mol% yttria. Adhesion between the abutment and crown is an important factor that influences the effectiveness and longevity of crown prostheses. According to Matsumura et al. [10], 10–20 MPa is the minimum clinically acceptable bonding strength for crown prostheses. Moreover, surface pretreatment is required for dental adhesives to obtain sufficient adhesive bonding strength [11,12,13]. It is known that chemical bonding between resins and materials such as metals and ceramics can be achieved. In particular, a sufficient bonding strength between metals and resins can be obtained by using adhesive monomers such as phosphate ester monomers and carboxylic acid monomers for non-precious metals, and sulfur-containing monomers for precious metals [14,15].

Surface pretreatment is required for adhesive bonding and the setting of zirconia prostheses. The surface pretreatment process generally requires a combination of sandblasting and the application of a primer containing 10-methacryloyloxydecyl dihydrogen phosphate (MDP) [16,17]. However, sandblasting causes microcracking by zirconia phase transformation and fracture of the zirconia crown margin. In addition, surface damage caused by sandblasting reduces the strength of zirconia. As a result, there is concern that the material will fail at stresses much lower than its ideal strength [18,19]. Therefore, a new surface treatment method that does not include sandblasting is required for zirconia.

In contrast, techniques such as metallization are used to coat a metal film on a nonmetal surface. Some studies have reported the metallization of zirconia surfaces using Ti and Au solders [20,21,22,23]. However, metallization using solders results in the nonuniform thickness of the metal film coating, thereby impairing the esthetics of zirconia; therefore, the clinical application of metallization using solders is limited. To overcome these challenges, uniformly thin metal films must be deposited by sputtering rather than metallization using solders. In addition, titanium and gold are metals that are actually used in dental treatment and will not cause any concerns related to biocompatibility [24,25]. However, few studies have reported the metallization of zirconia surfaces by sputtering. Therefore, we performed metallization of YSZ by sputtering and evaluated the bond strength between YSZ and resin cement. Furthermore, the adhesion between the metallized YSZ and resin cement was characterized using surface analysis techniques.

The purpose of this study was to investigate the possibility of metallization by sputtering as a new surface treatment method for YSZ and to improve the bond strength between YSZ and resin cement.

## 2. Materials and Methods

### 2.1. Materials

The materials used in this study are listed in Table 1. A schematic of the test setup is shown in Figure 1. A total of 24 fully sintered YSZ disks (AZ ONE, Osaka, Japan; specimen size: 12.0 mm in diameter and 5 mm in height, density: 5.4 g/cm^3^, porosity: <0.01%) and stainless-steel rods (Stainless Hikari, Yamaguchi, Japan; specimen size: 6.0 mm in diameter and 5 mm in height; SUS304) were used. One end of each YSZ disk and SUS304 rod was polished under tap water using a SiC waterproof abrasive paper in the order of #400, #600, #800, and #1200, and finally using a diamond abrasive paper (grain size: 15 μm). After polishing, the YSZ disks and SUS304 rods were ultrasonically cleaned in acetone for 3 min and then air-dried.

### 2.2. Surface Treatment of YSZ

The polished YSZ specimens were classified into three groups according to the following surface treatment processes. Each experimental group had eight specimens (*n* = 8).

**Control group:** Polished YSZ was used as the control.**Au sputtering group:** Au was sputtered on the YSZ surface using an ion coater (IB-II, Eiko, Japan) for 15 min. After sputtering, the YSZ specimens were heated at 800 °C for 15 min in a porcelain furnace (JELENKO LT ‖, Morita, Osaka, Japan) in air, and then cooled to room temperature. After cooling, V-primer (Sun Medical, Shiga, Japan)—which enhances the bonding strength—was applied to the YSZ surface according to the manufacturer’s instructions.**Ti sputtering group:** Ti was sputtered on the YSZ surface using a high-frequency sputtering unit (SBR1104E, ULVAC, Kanagawa, Japan) at 150 W for 8 h. After sputtering, the YSZ disks and the specimens in the Au sputtering group were heated to 800 °C for 15 min in a porcelain furnace in air and were then cooled to room temperature.

### 2.3. Bonding Procedures

After the surface treatment, the YSZ disks were bonded to the end surface of the SUS304 rods using resin cement (Super-Bond C&B, Sun Medical, Shiga, Japan) according to the manufacturer’s instructions. Then, a constant load was applied to the SUS304 rods using a clamp to enable adhesion to the resin cement. Excess resin cement was removed from the bonding edge using a dental probe before curing. The bonding area was defined as the surface area of the ends of a SUS304 rod. According to the International Organization for Standardization’s recommendation, specimens were prepared at 23 ± 2 °C and stored in water at 37 ± 1 °C for 24 ± 1 h [26,27].

### 2.4. Measurement of Shear Bond Strength

Shear bond strength tests were performed at a crosshead speed of 1.0 mm/min using a universal testing machine (Autograph AG-1000E, Shimadzu, Kyoto, Japan) to obtain the maximum load [27,28]. The shear bond strength (S; MPa) was calculated using the following formula:S = P/A,
where P is the maximum load obtained from the test and A is the cross-sectional area of the SUS304 rod. In this study, the bonding area was 9π mm^2^.

### 2.5. Surface Analysis, Fracture Surface Observation, and Line Analysis

The Ti-sputtered YSZ surface was analyzed using X-ray photoelectron spectroscopy (XPS; Quantum 2000, ULVAC, Kanagawa, Japan) at 24 W before and after heat treatment to identify the chemical constituents and elemental states of the YSZ disks. The binding energies were calibrated by the C1s hydrocarbon peak at 285.0 eV. The XPS profiles were analyzed using the MultiPak software (ULVAC).

Secondary electron images (SEIs) were obtained using an electron probe microanalyzer (EPMA-1610, Shimadzu, Kyoto, Japan; EPMA) to observe the morphological characteristics of the fracture surface of the YSZ disks. Elemental mapping was performed on the fracture surfaces of the YSZ disks in the Au and Ti sputtering groups using EPMA.

For EPMA line analysis of the treated surface, a specimen in the Ti sputtering group was diagonally cut and polished, and its cross section was analyzed using EPMA at a voltage of 15 kV and a measurement time of 1.3 h.

### 2.6. Statistical Analysis

The statistical analysis was performed using the Excel 365 software (Microsoft, Redmond, WA, USA). The results of shear bond strength tests were statistically analyzed using one-way ANOVA followed by Tukey’s test (*p* < 0.05).

## 3. Results

### 3.1. Shear Bond Strength

Figure 2 shows the results of shear bond strength tests. The mean shear bond strengths for the control, Au sputtering, and Ti sputtering groups were 12.5 ± 2.5, 11.8 ± 2.4, and 22.9 ± 4.4 MPa, respectively. The shear bond strength of the Ti sputtering group was significantly higher than that of the other groups.

### 3.2. XPS Analysis

Figure 3a1,a2 shows the deconvolution of the Ti2p and O1s peaks, respectively, for the Ti sputtering group before heating. Figure 3b1,b2 shows the deconvolution of the Ti2p and O1s peaks, respectively, of the specimen after heating. The binding energies and atomic concentrations are summarized in Table 2. The binding energies of Ti2p for different pretreatments with or without heating are plotted in the graph. The XPS peak shifts of Ti2p3/2 and O1s can be observed in the Ti sputtering group before and after heating. The atomic oxygen concentration increased after heating.

### 3.3. Fracture Surface Observation

The SEIs of the fracture surfaces after the shear bond tests are shown in Figure 4. The YSZ surface and resin cement were identified in every group. However, for the control group, the failure mode was almost the failure of adhesion between the resin cement and the YSZ surface. For the Au sputtering group, the predominant failure mode was the adhesion failure between the cement and the YSZ surface; however, cohesion failure was also observed. Moreover, for the Ti sputtering group, a cohesion failure mode in cement and a combined adhesion and cohesion failure mode were observed. Comparing the three groups, a larger amount of resin cement remained on the surface of the Ti sputtering group.

### 3.4. EPMA Elemental Mapping

Figure 5a,b shows the SEIs and the results of elemental mapping on the fracture surfaces for the Au- and Ti-sputtered specimens, respectively. The intensity of the Au signal detected on the fracture surface of the Au-sputtered specimen was weaker than that of the Zr signal, as shown in Figure 5a. Unlike the intensity of the Zr signal, the intensity of the Ti signal detected on the fracture surface of the Ti-sputtered specimen was strong, as shown in Figure 5b.

### 3.5. EPMA Line Analysis

Figure 6 shows the results of the EPMA line analysis of the Ti-sputtered specimen. The red line represents the signal intensity of oxygen, and the green line represents that of Ti. The area enclosed by the blue line indicates the integrated range of the line analysis. The signal intensity of oxygen rapidly decreased near the surface of the specimen. In contrast, a rapid increase in the signal intensity of Ti can be observed near the surface.

## 4. Discussion

For clinical applications, the minimum required bonding strength is 10 MPa. A bonding strength greater than 20 MPa is desirable [10,11,12,13]. In this study, only the Ti sputtering group exhibited a bond strength of over 20 MPa. Matsumura et al. [29] reported that the bonding strength between zirconia disks using a resin cement (Super-Bond C&B) after sandblasting was approximately 40 MPa. The mean bond strength for the Ti sputtering group was 21.5 ± 2.4 MPa. The mean value exceeded 20 MPa; however, it was lower than 40 MPa. This difference between the bonding strengths can be attributed to the anchor effect and the increased surface area for bonding owing to sandblasting. Moustafa et al. [30] and Zhang et al. [31] reported that sandblasting facilitates the fatigue fracture of YSZ. Therefore, although sandblasting treatment can improve the bonding strength, it may not be possible to predict the long-term performance of zirconia prostheses.

The SEI shows the polishing marks made during the sample preparation on the YSZ surface of the control group. This is due to the exposure of the YSZ surface caused by the destruction of the interface between the resin cement and the YSZ surface.

Similar abrasions were observed on the surface of the Au-sputtered specimens. However, no such marks were observed on the surface of the Ti-sputtered specimens. This suggests that the Au-sputtering layer peeled off the YSZ surface of the Au-sputtered specimens, whereas the Ti-sputtered layer remained on the YSZ surface.

The EPMA elemental mapping revealed that the intensity of the detected Au signal on the fracture surface of the specimen in the Au sputtering group was weaker than that of the Zr signal, whereas the intensity of the Ti signal was stronger than that of the Zr signal on the fracture surface of the specimen in the Ti sputtering group. Therefore, the Au-sputtering layer may partially adhere to the YSZ surface, and the Ti-sputtering layer may completely adhere to the surface. The sufficient bond strength between the YSZ surface and resin cement may be explained by the elemental characterization of the sputtered film, as described below.

The metallized layer of Ti was analyzed by XPS. Based on the reports of Briggs et al. [32] and Bharti et al. [33], it was possible to accurately determine the formation of sub-oxide Ti_2_O_3_ and TiO_2_ on the TiO_x_ film from the XPS fit. TiO_2_ and Ti_2_O_3_ were formed in the specimens before heating. However, only TiO_2_ was formed in the heated specimen. From the XPS binding energy, the peak of Ti_2_O_3_ shifted to TiO_2_. This indicated that the oxidation of Ti progressed upon heating. Ida et al. [34] reported that the oxidation of Ti progresses rapidly at approximately 800 °C. This indicates that a direct transformation of Ti_2_O_3_ into TiO_2_ occurred, and the atomic concentration of oxygen increased.

The thickness of the target material can be measured by depth profiling using XPS, but the measurement value is generally based on SiO_2_. In this measurement, the thickness was found to be 1 μm in the SiO_2_ equivalent, but it can be predicted to be much thinner.

According to the results of the EPMA line analysis, the diffusion of oxygen was confirmed for most of the surface layer of the YSZ specimens, whereas diffusion of Ti was not observed. Since the diameter of the EPMA detection probe was about 5 μm, the detected signal from the area with the metallized layer of Ti, which was less than 1 μm, was combined with the signal from the other area. In addition, the total signal intensity of a specific area is displayed here as a line analysis, so if the total signal intensity is small, it may appear as if the concentration of the element of interest has decreased. However, it can be seen that oxygen diffusion occurred between the titanium and zirconia. This suggests that the diffusion of oxygen is significant for the metallization of YSZ. From the references [21,23,24], it is considered difficult for titanium to diffuse under this study’s conditions, but titanium was added as an item to confirm that there is no diffusion of titanium.

Pimenta et al. [23] and Lin et al. [24] reported that the oxygen in zirconia combined with Ti when stabilized zirconia was bonded with Ti solder and heated. In our study, the same reaction was observed. Furthermore, few ZrO_2_ particles on the YSZ surface could reduce to the form ZrO_2−x_ (x < 2) owing to the diffusion of oxygen, and an oxide of TiO_x_–ZrO_2−x_ could be formed at the interface between the YSZ surface and the metallized layer. Therefore, the diffusion of oxygen at the interface between the YSZ and the sputtered Ti layer contributed to the formation of a strong metallized layer, consequently enhancing the shear bond strength between the YSZ and resin cement.

A new and effective surface treatment for YSZ was developed for metallization by Ti sputtering. This metallization technique for YSZ has potential in clinical applications.

## 5. Conclusions

In summary, within the limitations of this study, the mean bond strength of YSZ to resin cement after Au and Ti metallization of the YSZ surface was 11.8 ± 2.4 and 22.9 ± 4.4 MPa, respectively. Therefore, the Ti metallization of YSZ is effective in improving the shear bond strength between YSZ and resin cement.

## Figures and Tables

**Figure 1 jfb-12-00062-f001:**
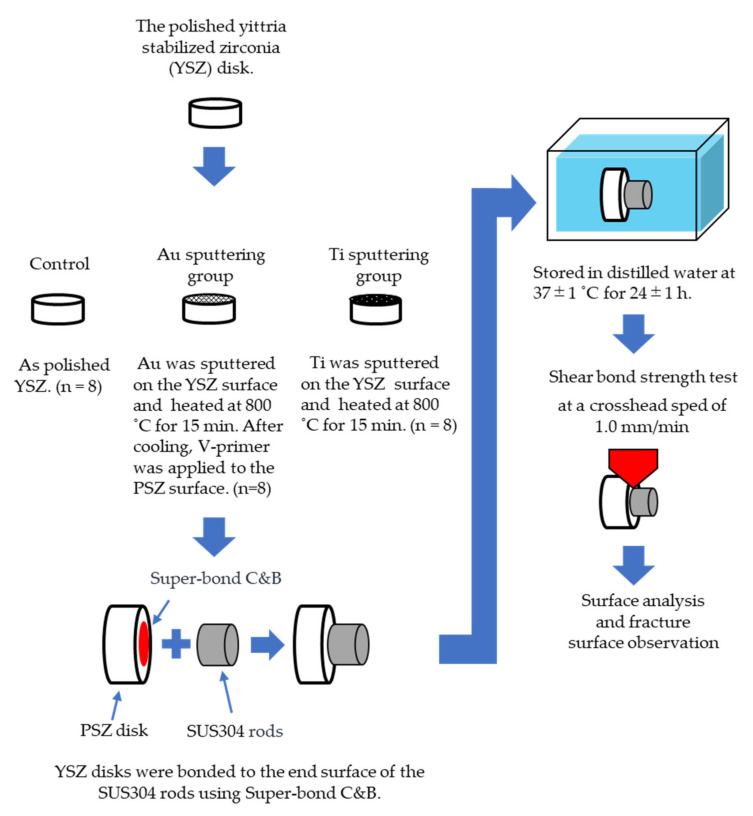
Schematic of the experimental procedure. Refer to Table 1 and the text for the abbreviation definitions.

**Figure 2 jfb-12-00062-f002:**
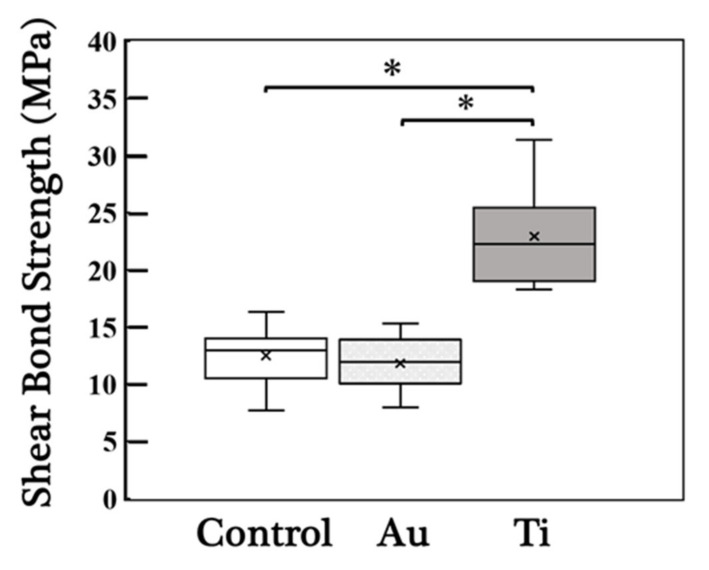
Box plot of the shear bond strengths. "×" indicates the average value. Au: Au sputtering group, Ti: Ti sputtering group. The shear bond strength of the Ti sputtering group was significantly higher than that of the other groups. Asterisks indicate *p* < 0.05 as compared with the values of the specimens. Bars denote standard deviation.

**Figure 3 jfb-12-00062-f003:**
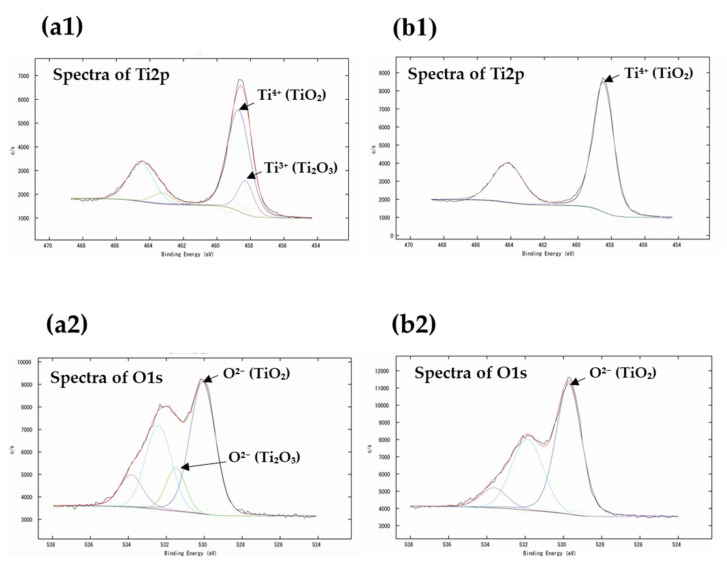
XPS profiles of the specimens: (**a1**,**a2**) Ti2p and O1s peaks before and (**b1**,**b2**) after heating of a specimen in the Ti sputtering group.

**Figure 4 jfb-12-00062-f004:**
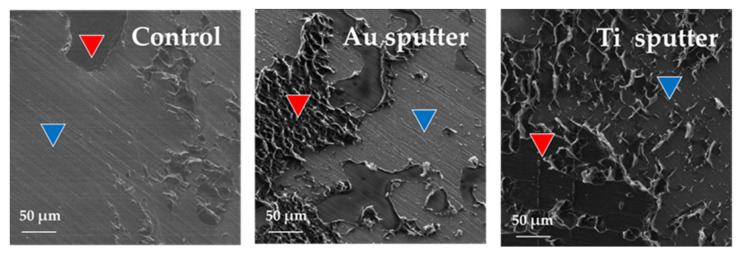
SEIs of fracture surface of the specimens. Red triangles indicate resin cement and blue triangles indicate YSZ surface.

**Figure 5 jfb-12-00062-f005:**
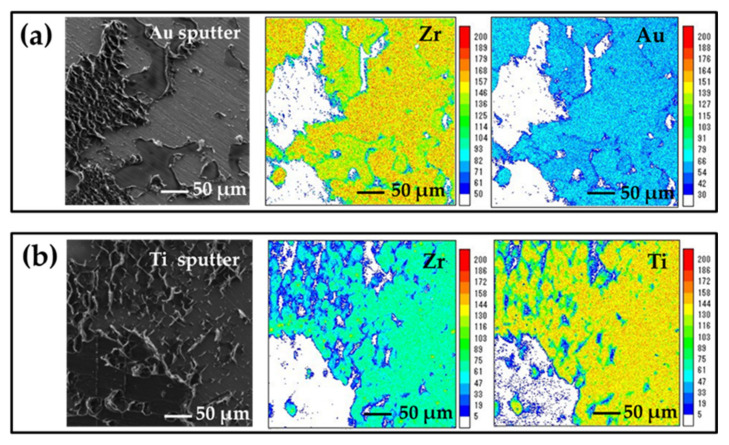
(**a**) SEI of the Au-sputtered specimen. Elemental mapping images of Zr and Au. (**b**) SEI of the Ti-sputtered specimen. Elemental mapping images of Zr and Ti.

**Figure 6 jfb-12-00062-f006:**
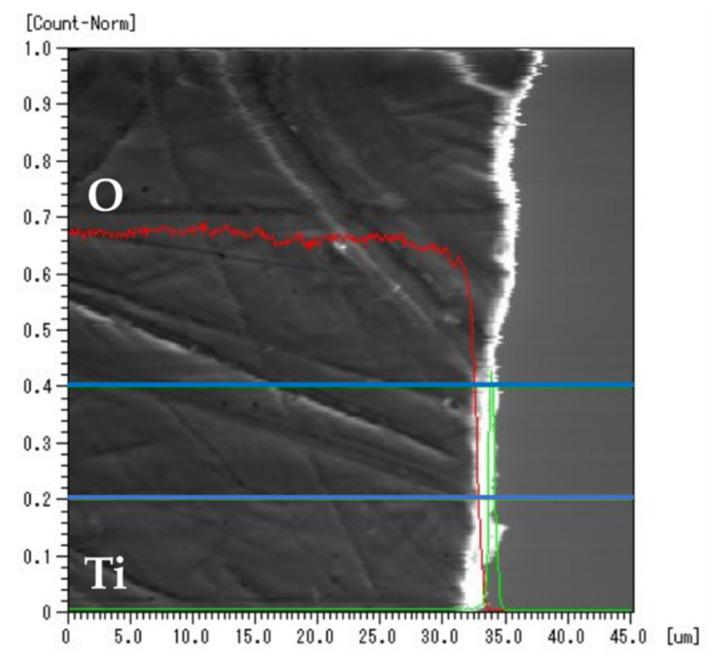
Line analysis of Ti-sputtered specimen after heating. The red and green lines denote the intensity of oxygen and Ti, respectively. The area enclosed by the blue line indicates the integrated range of the line analysis.

**Table 1 jfb-12-00062-t001:** Summary of materials used in this study.

Materials/Product Name	Abbreviation	Manufacturer	Composition
Yttria stabilized zirconia/ZR8Y-12.0	YSZ	AS ONE Co., Ltd.	92.0% ZrO_2_, 8.0% Y_2_O
Stainless steel/SM995	SUS306	Stainless Hikari Co., Ltd.	72.0% Fe, 8.0% Ni, 18.0% Cr, 2.0% others
Resin cement/Super Bond C&B	SB	SUN MEDICAL Co., Ltd. (Moriyama City, Japan)	4-META, TBB, PMMA, MMA
Adhesive monomer/V primer	VP	SUN MEDICAL Co., Ltd.	0.1 mol% VBTDT, acetone

4-META: 4-methacryloxyethyl trimellitate anhydride; TBB: tri-n-butyl borane; MMA: methyl methacrylate; PMMA: polymethyl methacrylate; VBATDT: 6-(4-vinylbenzyl-n-propyl) amino-1, 3, 5-trizaine-2,4-dithiol.

**Table 2 jfb-12-00062-t002:** XPS binding energies and atomic concentrations on the surface of a specimen in the Ti sputtering group before and after heating.

	XPS Binding Energy (eV)				Atomic Concentration (at%)			
	Ti2p		O1s		C1s	O1s	Ti2p	Zr3d
Before heating	458.7	458.3	531.5	530.1	60.19	33.63	6.10	0.09
After heating	458.5		529.7		53.39	39.06	7.55	0

## Data Availability

The data presented in this study are available on request from the corresponding author.

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
