# Peer review of "Metallization by Sputtering to Improve the Bond Strength between Zirconia Ceramics and Resin Cements"

_jfb, 2021, doi:10.3390/jfb12040062_

Round 1
Reviewer 1 Report
The manuscript is quite interesting and shows a very practical approach to the dental restoration issue. However it seems incomplete and there are some errors that need to be corrected.
- The authors used ZrO2 with 8 mol % of Y2O3 and call it “partially stabilized zirconia, PSZ”. It was assumed that authors wrote about additives in mol% because usually in this way such materials are described in literature. However that should be clarified. So, writing that this ceramics is PSZ is a huge mistake. Partially stabilized zirconia contains 3 or 5 mol % of Y2O3. If zirconia contains above 6-7 mol % of Y2O3 such material is considered as fully stabilized.
- There is no information about used zirconia materials. The authors should write if it was fully sintered or it was in “biscuit” state. What about density and porosity? Usually, the companies sells a partially sintered zirconia materials which after CNC processing are finally sintered and implemented as a prostheses materials. So, it is unclear if authors used a partially sintered materials and in this way porous materials or it was fully densified. If it was porous materials, a porosity might influence on bonding strength, similarly to sandblasted zirconia case.
- The authors should explain why the samples were immersed in water for 24 hours. Such short time has no effect on zirconia, and should not has effect on bonding resin.
- The authors should explain why the sputtering time in case of Ti was 8 h.
- The authors should write what was a thickness of sputtered layers.
- Why the EPMA line analysis was conducted on sputtered samples but not on heated samples? Admittedly, there is such information in figure 6 caption but it isn’t mentioned in methods, line analysis or discussion parts. There should be a clear information about it.
- Couldn't cutting and polishing cause additional Ti oxidation and in such a way influence on results?
- What does it means that the SEIs showed polished marks on the PSZ surface? It was understood that after a shear bond strength test there was no polishing or other treatments.
- The authors should explain what does it means that “the diffusion of oxygen was confirmed for most of the surface layer of the PSZ specimens, whereas diffusion of Ti was not observed”. What kind of diffusion the are talking about? If they used sintered zirconia, such material is rather inactive in 800° Of course there might be some ionic conductivity at elevated temperature in the fully stabilized zirconia, what is used in solid oxide fuel cells. However heating in air atmosphere caused rather Ti oxidation by an oxygen present in air. So, in such conditions, it is hard to expect that Ti would like to diffuse into zirconia.
Author Response
Thank you for your suggestion.
Please see the revised manuscript and my comment.
(1)The authors used ZrO2 with 8 mol % of Y2O3 and call it “partially stabilized zirconia, PSZ”. It was assumed that authors wrote about additives in mol% because usually in this way such materials are described in literature. However that should be clarified. So, writing that this ceramics is PSZ is a huge mistake. Partially stabilized zirconia contains 3 or 5 mol % of Y2O3. If zirconia contains above 6-7 mol % of Y2O3 such material is considered as fully stabilized.
(A) Thank you for your suggestion. Regarding your suggestion, we have corrected PSZ (partially stabilized zirconia),to YSZ (Yttria stabilized zirconia).
(2)There is no information about used zirconia materials. The authors should write if it was fully sintered or it was in “biscuit” state. What about density and porosity? Usually, the companies sells a partially sintered zirconia materials which after CNC processing are finally sintered and implemented as a prostheses materials. So, it is unclear if authors used a partially sintered materials and in this way porous materials or it was fully densified. If it was porous materials, a porosity might influence on bonding strength, similarly to sandblasted zirconia case.
(A) Zirconia used in this study is a fully sintered one, and I have mentioned that in the text. We asked the manufacturer about the density and porosity, but they said they could not disclose them because they were trade secrets.
(3)The authors should explain why the samples were immersed in water for 24 hours. Such short time has no effect on zirconia, and should not has effect on bonding resin.
(A)The purpose is to increase the degree of polymerization of the resin cement by postcuring and to check the initial bond strength. (This method is specified in ISO TR 11405). Since we were able to confirm the initial adhesion strength, we will consider the degradation test in future experiments.
(4)The authors should explain why the sputtering time in case of Ti was 8 h.
(A)We conducted preliminary experiments with processing times of 1, 2, 4, and 8 hours. It was able to form a stable thin film in 8 hours with this experimental machine.
(5)Why the EPMA line analysis was conducted on sputtered samples but not on heated samples? Admittedly, there is such information in figure 6 caption but it isn’t mentioned in methods, line analysis or discussion parts. There should be a clear information about it.
(A)Added the following details about line analysis to the Materials and Methods.
2.5. Surface Analysis, Fracture Surface Observation, and Line Analysis
The Ti-sputtered YSZ surface was analyzed using X-ray photoelectron spectros-copy (XPS; Quantum 2000, ULVAC, Kanagawa, Japan) at 24 W before and after heat treatment to identify the chemical constituents and elemental states of the YSZ disks. The binding energies were calibrated by the C1s hydrocarbon peak at 285.0 eV. The XPS profiles were analyzed using the MultiPak software (ULVAC).
Secondary electron images (SEIs) were obtained using an electron probe micro-analyzer (EPMA-1610, Shimadzu, Kyoto, Japan; EPMA) to observe the morphological characteristics of the fracture surface of the YSZ disks. Elemental mapping was per-formed on the fracture surfaces of the YSZ disks in the Au and Ti sputtering Groups using EPMA.
For EPMA line analysis of the treated surface, a specimen in the Ti sputtering Group was diagonally cut and polished, and its cross section was analyzed using EP-MA at a voltage of 15 kV and a measurement time of 1.3 h.
(6)Couldn't cutting and polishing cause additional Ti oxidation and in such a way influence on results?
(A)Since, the process is carried out at low speed and being cooled under water, there was no effect by heat.
(7)What does it means that the SEIs showed polished marks on the PSZ surface? It was understood that after a shear bond strength test there was no polishing or other treatments.
(A)It means that the polishing marks on the zirconia surface were observed in SEI when the zirconia surface was polished during specimen preparation. The text has been modified as follows.
SEIs showed polishing marks made during the preparation of the specimens on the YSZ surface in the control Group, which can result from the destruction of the interface between the resin cement and the YSZ surface, exposing the surface of YSZ.
(8)The authors should explain what does it means that “the diffusion of oxygen was confirmed for most of the surface layer of the PSZ specimens, whereas diffusion of Ti was not observed”. What kind of diffusion the are talking about? If they used sintered zirconia, such material is rather inactive in 800° Of course there might be some ionic conductivity at elevated temperature in the fully stabilized zirconia, what is used in solid oxide fuel cells. However heating in air atmosphere caused rather Ti oxidation by an oxygen present in air. So, in such conditions, it is hard to expect that Ti would like to diffuse into zirconia
(A)We think it is difficult to diffuse titanium for zirconia under the current conditions as your suggestion. It was necessary to confirm that no diffusion of titanium was observed, so it was added to the inspection items.
(9)The authors should write what was a thickness of sputtered layers.
(A) XPS can measure the thickness of the target material, but the measured value is based on the SiOâ‚‚ thickness. In this measurement, the thickness was found to be 1 μm in SiOâ‚‚ equivalent.
Reviewer 2 Report
Dear Author,
small, but enough well written and interesting manuscript. However, I have some objections regarding the:
1) Introduction - too short. Please, expand the functional significance of zirconia in clinical usage more. This is interesting for the reader and you have to provide more info in comparison with described now. Additionally, this will add also References to you quite short Reference list!
2) materials and methods. Please, give basis, why used in the Table 1 materials were used. there is not enough only with the Fig. 1 scheme, please give short description and abbreviations beneath the Fig. Refer the Reference where it is possible, for the experimental procedure and also for the 2.2 and 2.4.
3) Discussion. Too short. Please give comparison/analysis of your data in relation to other author data. Mandatory paragraph ir Limitations for the research.
4) Conclusions. OK, but develop them in the Present time.
5) References. Highly not enough for this manuscript and this Journal. You will increase the number by expansion of Introduction and Discussion. Well, there are 2 old references from the previous century. Please, re-think the necessity of them and try perhaps to remove or to replace with the other ones.
Author Response
Dear reviewer
Thank you for your suggestion.
I will send you the answer. Please check it in the revised manuscript.
1) Introduction - too short. Please, expand the functional significance of zirconia in clinical usage more. This is interesting for the reader and you have to provide more info in comparison with described now. Additionally, this will add also References to you quite short Reference list!
A)Reference [2-6,8,9,11-16] added and appended to the introduction.
2) materials and methods. Please, give basis, why used in the Table 1 materials were used. there is not enough only with the Fig. 1 scheme, please give short description and abbreviations beneath the Fig. Refer the Reference where it is possible, for the experimental procedure and also for the 2.2 and 2.4.
A) Modified Figure 1 to add reference [26,27] to Materials and Methods.
3) Discussion. Too short. Please give comparison/analysis of your data in relation to other author data. Mandatory paragraph ir Limitations for the research.
A)Added Limitations statement and Discussion.
4) Conclusions. OK, but develop them in the Present time.
A)Corrected to present tense
5) References. Highly not enough for this manuscript and this Journal. You will increase the number by expansion of Introduction and Discussion. Well, there are 2 old references from the previous century. Please, re-think the necessity of them and try perhaps to remove or to replace with the other ones.
A) Added more references. The old reference Narita et al.[9] has been replaced by the new reference Xia et al. [21]. However, reference Ida et al. [32] has been retained as it is an essential part of the text as a reference.
Reviewer 3 Report
This manuscript describes an interesting study of metallization by sputtering as a new surface treatment method to improve the bond strength between Zirconia Ceramics and Resin Cements. The description of the experiments are understandable and easily reproducible. The images are very clear and well described. The statistical analysis of the experimental data is done well. The references are few and not recent.
Author Response
Dear reviewer
Thank you your suggestion.
Added more references.
Reviewer 4 Report
The manuscript titled “Metallization by Sputtering to Improve the Bond Strength between Zirconia Ceramics and Resin Cements” by Kimura et al is a well-written research paper mainly describes a surface modification strategy to overcome the microcracking on zirconia introduced during sandblasting by conducting metallization using Ti and Au sputtering and heat treatment. Ti sputtered zirconia demonstrated enhanced shear bonding strength. The failure mechanism during Instron Test has also been revealed by observing the SEI morphologies of the samples. The overall flow of the manuscript is clear but seems too simple. The experimental design also lacks rigor. My enthusiasm to this paper is relatively low.
- The aim of this study is trying to diminish the chips and cracks of zirconia surface produced during sandblasting. However, the authors did not even show the surface morphologies after metallization.
- Normally, sandblasting is only the initial pretreatment of surface modification to create a rough surface. And other surface modification techniques such as acid etching, alkaline treatment, sputtering, ion implantation, etc. are followed. During the follow up treatment, most chips and cracks will be gone. The authors aim to find a method to get rid of sandblasting, I think the motivation here does not carry conviction.
- The authors claims that the proposed metallization technique for PSZ has potential for clinical applications. However, neither biocompatibility nor bioactivity has been evaluated on the modified surfaces.
- The sputtering parameters for Au and Ti are totally different. Too many variables make the study out of control: sputtering elements (Au/Ti), methods (ion coater/ high-frequency sputtering unit), time (15 min/8 h). The current experimental design cannot reveal the reason why Ti sputtered surface exhibited better mechanical property.
- The microstructures (surface, cross-section) of both sputtered samples are expected to be provided and the relationship between the microstructures and performances (shear strength or even biocompatibility) are also expected to be discussed.
Author Response
Dear reviewer
Thank you for your suggestion.
I'll send you my answer.
(1)The aim of this study is trying to diminish the chips and cracks of zirconia surface produced during sandblasting. However, the authors did not even show the surface morphologies after metallization.
(A)The phase transition point of zirconia is 1170°C.The heat given to zirconia during the heat treatment and RF sputtering in this study is lower than this temperature, so we considered that damage to zirconia is unlikely to occur.
(2)Normally, sandblasting is only the initial pretreatment of surface modification to create a rough surface. And other surface modification techniques such as acid etching, alkaline treatment, sputtering, ion implantation, etc. are followed. During the follow up treatment, most chips and cracks will be gone. The authors aim to find a method to get rid of sandblasting, I think the motivation here does not carry conviction.
(A)As shown in Moustafa et al. [28] and Zhang et al. [29], zirconia is damaged just by sandblasting, so I think that the damage will not disappear even if we perform a treatment like this one after the sandblasting.
(3)The authors claims that the proposed metallization technique for PSZ has potential for clinical applications. However, neither biocompatibility nor bioactivity has been evaluated on the modified surfaces.
(A)The materials used this study are a combination of materials used in living organisms. We considered that the problem with biocompatibility and bioactivity are unlikely to occur.
(4)The microstructures (surface, cross-section) of both sputtered samples are expected to be provided and the relationship between the microstructures and performances (shear strength or even biocompatibility) are also expected to be discussed.
(A)From the results of the surface observation, there was an interface failuere between Au and zirconia, and from the results of the shear bond test, the Au sputtering group could not provide the bonding strength. From these results, we judged that the metallization layer could not be formed and did not perform a line analysis.
(5)The sputtering parameters for Au and Ti are totally different. Too many variables make the study out of control: sputtering elements (Au/Ti), methods (ion coater/ high-frequency sputtering unit), time (15 min/8 h). The current experimental design cannot reveal the reason why Ti sputtered surface exhibited better mechanical property.
(A)Each machine supports only one of Ti and Au, we change the machine depending on the target.
For Au, we followed the time recommended by the manufacturer.
However, there was no data on the film thickness of Ti sputtering machines, although the principle is the same, so preliminary experiments were conducted. 8 hours was optimal for the formation of a film similar to that of Au.
Round 2
Reviewer 1 Report
Regarding the authors' responses to previous comments:
Ad 1) explanation approved
Ad 2) One of the comments to the article was that the authors did not provide information about YSZ disks. The authors claims that they asked the manufacturer about the density and porosity, but they said they could not disclose them because they were trade secrets. Measurements of density and porosity by Archimedes methods is a basic kind of measurement and any laboratory with analytical balance can do it by themselves. So there is no reason to ask the manufacturer. Thus the authors explanations are insufficient. The authors should provide these information.
Ad 3) The authors explained to reviewer but it is not understood why they do not put this information in text. Please, explain this in text. ( refer to standard)
Ad4 ) explanation approved
Ad 5) explanation approved
Ad 6) explanation approved
Ad 7) The explanation and text modification is not sufficient. The authors wrote that they observed polishing marks (marks made during polishing) but in the same sentence they wrote that this marks can result from the destruction of the interface between the resin cement and the YSZ surface. I agree only with the first part of sentence. Pleas correct this and write it clearly and understandable.
Ad 8) explanation approved but the authors should put some more clear explanations in the text
Ad 9) Explanation completely unsatisfactory. The authors have to show (prove) the thickness of sputtered layer before and after heating (SEM or TEM images). Also, they should present how it looks like after the shear bond strength tests.
Moreover, the Figure 6 should be improved. It is unclear if this white line is a TiO2 layer or it is just a bad contrast and artifact connected with edge effects. So where is this layer?
What is more, the authors wrote that after heating there is TiO2 on the surface, so why in Fig 6. the signal of oxygen drops to 0 and just after that the signal from Ti is observed? These two signals should covered themselves
One more thing. What the authors had on mind when they wrote that “This suggests that the diffusion of oxygen is significant for the metallization of YSZ” (line 227-228 )? First of all they should talk rather about oxidation of Ti not diffusion of oxygen. Metallization was made during sputtering, so oxygen diffusion or oxidation in post treatment have nothing with this process. Thus this part in discussion is completely unclear.
Generally, the author’s explanations and discussions of the results are short and very often not unsatisfactory. They should describe and explain their work more widely
Author Response
Thank you for pointing this out.
Ad2) I measured the density and porosity as you suggested, and added them to the Materials and Methods section.
Ad 3) Added references and put this information in the materials and methods section.
Ad 7) I rewrote the discussion section as follows.
The SEI shows the polishing marks made during sample preparation on the YSZ surface of the control group. It would due to the exposure of the YSZ surface caused by the destruction of the interface between the resin cement and the YSZ surface.
Ad 8&9 ) As shown in the references, the industrial metallizing method is performed by heating, and we think that sputtering alone t is not sufficient for metallizing.
Therefore, we consider heating and oxidation to be metallization.
Regarding the white line part in Fig. 6, it is considered that a part is the TiOâ‚‚ layer, but it is difficult to distinguish it from the white part of the edge effect, and it is difficult to measure the thickness from this figure.
Also, theoretically, the two signals should cover each other as you mentioned. However, in actual measurement, it is difficult to get the desired figure because the integrated values are within the range.
Since the metallization layer is less than 1 μm as I mentioned last time, and the diameter of the EPMA detection probe used in this study is about 5 μm, the detected signals from the area below that will be merged with the signals from other areas. Furthermore, since the total signal intensity of a certain area is displayed here as a line analysis, if the total signal intensity is small, it may be displayed as if the concentration of the element of interest decreased.
Based on these, I added the following to the discussion.
The thickness of the target material can be measured by depth profiling using XPS, but the measurement value is generally based on SiOâ‚‚. In this measurement, the thickness was found to be 1 μm in SiOâ‚‚ equivalent, but it can be predicted to be much thinner.
According to the results of EPMA line analysis, the diffusion of oxygen was confirmed for most of the surface layer of the YSZ specimens, whereas diffusion of Ti was not observed. Since the diameter of the EPMA detection probe is about 5 μm, the detected signal from the area with the metallized layer of Ti which less than 1 μm will be combined with the signal from the other area. In addition, the total signal intensity of a specific area is displayed here as a line analysis, so if the total signal intensity is small, it may be displayed as if the concentration of the element of interest has decreased. However, it can be seen that oxygen diffusion is occurring between titanium and zirconia. It suggests that the diffusion of oxygen is significant for the metallization of YSZ. From the references[21, 23, 24], it is considered difficult for titanium to diffuse under this study conditions, but titanium was added as an item to confirm that there is no diffusion of titanium.
Reviewer 2 Report
Dear Authors,
despite you addressed almost all what was asked from you, there is a Limitation paragraph, what is absent. Limitation means to indicate the possible minuses of the research what relates to it (some additional method that could be used, but was not possible, some problems with material etc...). Please, remember for the next time, that Limitation paragraph is not a statement, but objective things what were not available in your research...
Author Response
Thank you for your advice.
I will be careful from the next time.
Reviewer 4 Report
The revised version had some significant improvement. However, there are still some issues that have not been appropriately addressed.
According to the explanation from the authors regarding the motivation of the current study, in introduction, it is recommended to discuss some other surface modification techniques in addition to sandblasting and talk some flaws of each technique. In this case, it will not confuse readers that the current work is specifically trying to diminish sandblasting. The technique proposed in this paper is neither an improvement, nor a substitution of sandblasting.
If the authors believe the material and surface modification treatment used in this study won’t cause any concern on cytotoxicity, it is recommended to include some references in this regard.
It is not surprised to be failed to test the bonding strength of Au coated sample in Instron. The thickness of the Au layer generated via a sputtering coater is probably only ~10 nm or slightly more. The sputtering pressure is also very low. So, it won’t give you a nice film or coating with good quality. I don’t think it is a surface modification method and I suggest removing the Au group throughout the whole manuscript. It is meaningless.
Author Response
Thank you for your suggestion.
(1)The description of sandblasting in the introduction has been added as follows.
However, sandblasting causes microcracking by zirconia phase transformation and fracture of the zirconia crown margin. In addition, surface damage caused by sandblasting reduces the strength of zirconia. As a result, there is concern that the material will fail at stresses much lower than its ideal strength [19,20]. Therefore, a new surface treatment method that does not include sandblasting is required for zirconia.
(2)I added a reference to biosafety and added the following to the introduction.
To overcome these challenges, uniformly thin metal films require to be deposited by sputtering rather than metallization using solders. In addition, titanium and gold are metals that are actually used in the dental treatment, won’t cause any concern on biocompatibility [25,26]
(3)Thank you very much for pointing this out.
As mentioned in the text, we chose gold as a precious metal because it can be metallized with gold solder, and from the viewpoint of biocompatibility, as well as against titanium, which is a non-precious metal.
It is unfortunate that we could not obtain sufficient results with gold, but we think it's necessary for comparative data.